# Systematic Review on the Link between Sleep Bruxism and Systemic Chronic Inflammation

**DOI:** 10.3390/brainsci13071104

**Published:** 2023-07-21

**Authors:** Michal Fulek, Mieszko Wieckiewicz, Anna Szymanska-Chabowska, Monika Michalek-Zrabkowska, Katarzyna Fulek, Gabriella Lachowicz, Rafal Poreba, Grzegorz Mazur, Helena Martynowicz

**Affiliations:** 1Department of Internal Medicine, Occupational Diseases, Hypertension and Clinical Oncology, Wroclaw Medical University, 213 Borowska St., 50-556 Wroclaw, Poland; michal.fulek@student.umw.edu.pl (M.F.); anna.szymanska-chabowska@umw.edu.pl (A.S.-C.); monika.michalek-zrabkowska@umw.edu.pl (M.M.-Z.); gabriella.lachowicz@umw.edu.pl (G.L.); rafal.poreba@umw.edu.pl (R.P.); grzegorz.mazur@umw.edu.pl (G.M.); 2Department of Experimental Dentistry, Wroclaw Medical University, 26 Krakowska St., 50-425 Wroclaw, Poland; m.wieckiewicz@onet.pl; 3Department and Clinic of Otolaryngology, Head and Neck Surgery, Wroclaw Medical University, 213 Borowska St., 50-556 Wroclaw, Poland; fulek.kasia@gmail.com

**Keywords:** sleep bruxism, inflammation, RMMA, systemic inflammation, teeth grinding, inflammatory, sleep

## Abstract

Sleep bruxism (SB) is a sleep-related behavior characterized as rhythmic (phasic) or non-rhythmic (tonic) masticatory muscle activity. SB is a common sleep behavior with a predominantly central origin. The aim of this systematic review was to evaluate the relationship between inflammatory status and SB according to the Preferred Reporting Items for Systematic Reviews and Meta-Analyses 2020 (PRISMA 2020). The research was registered at PROSPERO (CRD42023395985). We performed a systematic literature analysis using five different databases. Furthermore, the backward snowballing technique was applied to identify additional papers. Initially, 28 papers were screened from the database search, and 162 papers were revealed in the backward snowballing process. Eventually, five articles were included. Data concerning the inflammatory status of patients experiencing SB were investigated and summarized. Due to the heterogeneity of the compared studies, only a qualitative comparison and narrative summary were performed. The results suggest that SB could be associated with systemic inflammation. In fact, this systematic review revealed that there are no papers conclusively showing that the inflammatory status in bruxers is comparable to non-bruxers. However, each of the examined studies utilized different methods of assessing systemic inflammation, which makes the results dubious.

## 1. Introduction

Sleep bruxism (SB) appears to be one of the main fields of interest among researchers involved in sleep medicine [1]. The SB construct has shifted from pathology to motor activity, with a possible physiological or even protective relevance [2]. Therefore, SB should no longer be regarded as a parafunctional act of grinding the teeth, leading to tooth wear or other damage [3,4]. The widely adopted definition of bruxism was first introduced in 2013 [5]. It has since been updated by Lobbezoo et al. in 2018 [6]. The definition states that SB is a rhythmic (phasic) or non-rhythmic (tonic) masticatory muscle activity during sleep. The definition emphasizes that it is not a movement disorder or a sleep disorder in otherwise healthy subjects [6]. Clinical symptoms of SB are classified in the third edition of the International Classification of Sleep Disorders and involve regular or frequent bruxism events during sleep and the presence of abnormal tooth wear or incidents of jaw muscle pain or fatigue [7]. Most of the theories surrounding the etiology of bruxism are focused on genetic and psychologic vulnerability to stress or anxiety [8,9], the role of certain neurotransmitters (serotonin [10], dopamine, gamma aminobutyric acid (GABA) and noradrenaline [11]), genetic origin [12,13,14], autonomic nervous system modulation [15] and exposure to exogenous risk factors, e.g., tobacco, alcohol or drugs and comorbidities [16,17]. It has recently been suggested that SB could be connected to systemic inflammation [18]. Even though short-lasting elevations in inflammation are crucial during physical strain, injury and infection, recent evidence [19] suggests that environmental and lifestyle factors can promote systemic chronic inflammation (SCI). SCI has been proven to lead to several diseases, including certain cancers [20] and type 2 diabetes, and has been found to increase the risk of developing cardiovascular events [21], ultimately reducing the duration of healthy life years (HLY) [22,23]. The prevalence of SB in the adult population is estimated to be between 8% [24,25] and 13% [26]. However, this value varies depending on the age group. SB is successfully established to escalate the risk of endothelial remodeling [27] and hypertension [28]. The necessity of finding common features in the patomechanism between SCI and SB is emphasized by the increase in cardiovascular risk accompanied by SCI [29]. Current ways of coping with SB focus only on symptoms rather than the etiology of SB. Understanding the links between SCI and SB can carve new space for new drug development. Up to this point, there has not been a proven causal relationship between SB and SCI. We still lack evidence to propose cause to consequence ratiocinate and therefore, given the high prevalence and potentially significant consequences of SB coexisting with systemic inflammation, we aimed to establish the current state of knowledge on this relationship.

## 2. Objectives

The main objective of the present systematic review was to investigate the current arguments on the relationship between SB and inflammatory markers according to the updated version (PRISMA 2020 [30]) of the Preferred Reporting Items for Systematic Reviews and Meta-Analyses (PRISMA). We focused on the critical revision of evidence for the systemic inflammatory status in the available original articles in the field of SB. The PICo question (modified version for qualitative questions) is as follows: P (Population)—Patients experiencing SB, I (Interest)—Systemic inflammation and Context (Co)—The general adult population. The authors also intended to evaluate the strengths and weaknesses of the reviewed reports, as well as to develop a new approach for assessing the consequences of SB.

## 3. Materials and Methods

This study followed the PRISMA 2020 statement guidelines [30] and was registered at PROSPERO under the number CRD42023395985.

### 3.1. Search Strategy

A systematic search of relevant studies published since the inception of the databases, all the way up to February 2023, was performed using the EBSCOhost search engine. The following databases were screened: MEDLINE, Health Source—Consumer Edition, Health Source: Nursing/Academic Edition, Academic Search Ultimate and ERIC. To minimize bias, two researchers (M.F. and M.M.-Z.) independently performed an online search for peer-reviewed papers using the following combination of keywords: (‘sleep bruxism’ OR ‘bruxing’ OR ‘bruxer*’) AND (‘systemic inflammation’ OR ‘inflammation*’ OR ‘inflammatory response’ OR ‘IL-1’ OR ‘Interleukin-1’ OR ‘Il-6’ OR ‘Interleukin-6’ OR ‘Il-8’ OR ‘Interleukin-8’ OR ‘cytokine*’ OR ‘CRP’ OR ‘C-reactive protein’ OR ‘TNF-alpha’ OR ‘TNF-α’ OR ‘fibrinogen’). This strategy yielded 28 records (see Figure 1; PRISMA flow diagram). After the automatic exclusion of non-English articles and the removal of duplicates, 26 papers were assessed, of which 4 were included in this analysis. Studies were included if the keywords were present in either the title, the abstract or the original keywords of the paper.

Additionally, the reference lists of selected studies and the papers that cited selected articles were screened to identify relevant articles. This method resulted in identifying a total of 190 papers. Next, two independent researchers (M.F. and M.M.-Z.) read abstracts, keywords and titles in order to establish if the paper reported an original study accounting for the associations between SB and inflammatory markers. Studies that one researcher positively qualified were left to the decision of the third investigator (K.F.). In case the abstract did not provide sufficient information to determine if the paper should be excluded, the researchers followed with reading the full text. This strategy resulted in excluding reviews, qualitative research and quantitative studies that mentioned either inflammation or SB but did not assess these constructs. Next, five full texts were assessed for eligibility and eventually included and analyzed in the systematic review.

### 3.2. Eligibility Criteria

Details of the selection processes are shown in Figure 1. Overall, the selection process was aimed at identifying any original studies determining the association between SB and inflammatory markers. SB could be assessed by all medical methods, starting from an interview or appropriate questionnaire, dental examination and/or instrumental examination including polysomnography. The following exclusion criteria were taken into consideration: non-English records, reviews, case reports, guidelines, book chapters and no available access to the full text. Finally, five manuscripts were included in the systematic review (see Table 1).

### 3.3. Data Extraction and Quality Assessment Procedures

Data extraction (see Table 1) was conducted independently by two researchers (M.F. and M.M.-Z.). Extracted data included details on SB and the inflammatory status, sample characteristics and the main findings of the original study. Selected statistical information and data necessary to conduct the quality evaluation were also retrieved. Any discrepancies identified during the process of data extraction and quality evaluation were resolved with a consensus method [35] involving discussions between two researchers (M.F., M.M.-Z.), with the intervention of a third researcher (K.F.). In particular, in case of a discrepancy between the two researchers (M.F. and M.M.-Z.), the third researcher (K.F.) retrieved respective data, conducted the quality evaluation independently and led the discussion aiming at reaching a consensus.

### 3.4. Risk of Bias Assessment

To evaluate the quality of the identified studies, a tool described by Kmet et al. [36] was applied separately for qualitative [18] and quantitative studies [31,32,33,34]. This instrument for quality assessment takes into consideration the following criteria: the precision of the research aims; the report of the study design, materials and methods; and the justification of the study’s relevance, sampling strategy and reflexivity. Each component was rated using a 3-point response scale (2 points for ‘yes’, 1 point for ‘partial’, 0 points for ‘no’). If the criterion was not applicable for a study, then its score was excluded from the computation of the overall score. The cut-off point for the inclusion was 55% (indicating a liberal quality) of the potential maximum score. The 55% threshold was chosen from five possible cut-off points (75, 70, 65, 60 and 55%) proposed by Kmet et al. [36], who defined cut-offs as ranging from conservative (75%) to liberal (55%). Overall, the quality of the five selected studies was evaluated; all of the studies met the 55% threshold and were included in the analyses. The results of the risk of bias assessment are presented in Figure 2 and Figure 3.

## 4. Results

The results of the search are displayed in Figure 1. The research revealed five articles, including 307 participants—191 patients and 116 controls. Four studies were quantitative studies and one study was a qualitative study. The characteristic of the included studies is described in Figure 1.

### Search Results and a Synthesis of Findings from Studies Included into the Systematic Review

The studies focused on numerous inflammatory parameters, which made it difficult to have a head-to-head comparison of the results. Furthermore, only three of the studies focused on inflammatory markers measured from blood [18,32,33]. Combined with the small number of research papers in this field, this made it impossible to perform a meta-analysis.

A study performed by Louca Jounger, S. et al. on 20 women with temporomandibular disorders (TMD) and well-matched controls investigated the response of cytokine levels locally in the masseter muscle, in experimental tooth clenching [33]. This may be an indirect reflection of systemic inflammation [37]. The patients enrolled in the study suffered from temporomandibular disorders, not bruxism itself [33]. The manifestation of clenching in patients with temporomandibular disorders (TMD) can, however, be a reflection of bruxism [38]. Interleukin 6 (IL-6) and interleukin 8 (IL-8) levels increased in response to tooth clenching in both groups, while interleukin 7 (IL-7), interleukin 13 (IL-13) and tumor necrosis factor (TNF) rose only in patients with TMD-related myalgia. Microdialysis revealed that the levels of IL-6, IL-7, IL-8 and IL-13 were higher in patients with TMD-related myalgia than controls [33]. Lago-Rizzardi, C. et al. enrolled 24 women with chronic facial musculoskeletal pain (CFMP), with a mean age of 46 years, in their study. The study aimed to establish blood parameters associated with stress in patients with facial musculoskeletal pain. Bruxism appeared to be one of the most frequently occurring complaints among the tested subjects—62.5% in the study group vs. 0.0% in the controls (*p* < 0.001). The investigation revealed that CRP, IgE, C3 and C4 concentrations, as well as the level of leukocytes, tended to be higher in the group of women with CFMP, as opposed to the control group [32]. However, it is worth mentioning that the values were not statistically different.

The study performed by Keskinruzgar et al. observed inflammatory and neurodegenerative processes using optical coherence tomography (OCT) [34]. The measurements of the retinal nerve fiber layer (RNFL), inner plexiform layer (IPL) and ganglion cell layer (GCL) in the bruxism group were seen to be significantly lower than in the control group (*p* < 0.05), which might be an indirect reflection of the neuroinflammatory process [39]. However, the choroidal thickness was significantly higher [34]. Kara et al., however, did not refer to the inflammatory status itself, but determined TOS (total oxidant status) [31], which may reflect systemic inflammation [40]. Elevated levels of TOS have been observed in patients with psychiatric disorders [31]. SB patients in this study also exhibited a higher total oxidant status (*p* < 0.05) as well as a significantly higher oxidative stress index (*p* < 0.001) [31].

The results of the research carried out by Michałek-Zrąbkowska et al. showed that the bruxism episode index (BEI) positively correlated with the concentrations of C-reactive protein [18]. Interestingly, a statistically significant difference was observed in the BEI between patients with normal CRP values and those with increased values. The mean fibrinogen concentration in the blood serum of the patients with severe SB (BEI > 4) was significantly increased to 2.86 ± 0.69 g/L, compared to those without SB (2.58 ± 0.45 g/L; *p* = 0.046). Serum fibrinogen levels were also positively correlated with BEI [18]. What is undoubtedly worth mentioning is that this is the only study out of those included that diagnosed SB, following the AASM standards and basing diagnoses on polysomnographic examination (which remains the gold standard tool in diagnosing SB) [41].

The search did not reveal any papers suggesting/proving that the inflammatory status in patients with SB is similar to what we can find in a control group. Together, the findings of the presented studies can be regarded as an indicator of the potential link between SB and systemic inflammation. Within the next few years, systemic inflammation is likely to become an important field of interest in the investigation of SB.

## 5. Discussion

### 5.1. Inflammatory Status

Low-grade inflammation is a declared cause of many noncommunicable diseases, including cardiovascular disease (CVD), type 2 diabetes and some cancers [29]. Recent findings support the thesis that SCI could be an accelerant in aging, often described as “inflammageing” [42]. It contributes to immunosenescence, sarcopenia and a reduction in healthy life expectancy [20,43]. A systematic review with a meta-analysis performed by Ye, Z., Hu, T. et al. [44] indicated that the systemic immune-inflammation index (SII) could be a potential biomarker for CVD development. Many studies have shown a strong correlation between immune and inflammatory responses in the development of atherosclerosis [45,46], which, in turn, is a major cause of a stroke and ischemic heart disease [47]. SCI has also been proven to be capable of contributing to the pathogenesis of the most common neurodegenerative diseases, and to the leading cause of dementia worldwide, which remains as Alzheimer’s Disease (AD) [48,49]. SCI has the potential of inducing a neuroinflammatory cascade in the brain and triggering Alzheimer’s Disease pathology [50]. There is substantial evidence supporting the hypothesis that CVD may contribute to AD progression. Inflammation has been identified as a core component in the pathogenesis of both AD and CVD. It has been proven to not only be a consequence but also a critical contributor to the course of these diseases [51]. Recent studies suggest that dental screening is necessary for patients with one of the most prevalent CVD diseases, namely arterial hypertension [28]. This particularly concerns patients presenting with symptoms of SB. Still, many questions remain unanswered, unveiling a strenuous area of research that necessitates further exploration. One may only wonder whether we will be able to fight systemic inflammation and in that way cure, or even prevent, diseases, which up to this point have only been managed symptomatically [52,53].

### 5.2. Bruxism Diagnosis

The consensus proposed in the RDC/TMD Consortium Network Bruxism Consensus Meeting (“Assessment of Bruxism Status”) on 20 March 2017 detailed a grading system for categorizing bruxism as follows: “possible”—rating based on a positive interview, “probable”—rating based on a clinical examination with or without a positive interview and “definite”—rating based on a positive instrumental examination with or without a positive interview and/or positive clinical inspection [6]. The method of diagnosing SB in the investigated papers was suboptimal and in four out of five articles it was not categorized consistently with the consensus proposed by Lobbezoo et al. in 2018 [6]. Most of the revised papers investigated the inflammatory status accompanied by bruxism and did not focus on bruxism itself. In fact, only Michalek-Zrabkowska et al. stated whether the enrolled subjects had possible, probable or definite bruxism [18]. In the study, 74 individuals with probable SB were subjected to single-night polysomnographic examination, meaning that the final diagnosis of the involved subjects was definite bruxism [18]. Louca Jounger et al. [33] conducted a study whereby subjects performed a 20 min tooth-clenching task imitating possible bruxism, which was based on self-report only. The unargued limitation of this study is emphasized by some papers suggesting that each sign or symptom of SB would represent different aspects of the motor activity of the jaw during sleep [54]. Therefore, the need to discriminate between the different bruxism activities and its possible clinical manifestations is affirmed [55,56]. Raphael, K. et al. had detected in a case–control study a high sustained muscle activity throughout the sleep period in 72% of the cases of patients with TMD [57]. However, the OPPERA prospective study found an inconsistent association between SB, awake bruxism and TMD [58]. Recent systematic reviews continue to report an overlap between bruxism, TMD and headache [59]. In short, the relationship between bruxism and TMD is not clear, requiring longitudinal epidemiological studies and experimental clinical data to reject the cause–effect relationship.

In another study, also focusing on self-reported bruxism, Lago-Rizzardi et al. [32] observed patients with chronic facial musculoskeletal pain. Bruxism was one of the accompanying symptoms. In this case, bruxism could not be ranked as more certain than ‘probable’. Keskinruzgar et al. [34] investigated inflammatory processes in subjects with possible bruxism. In this study, patients were understood to be those experiencing teeth grinding and/or clenching at least five times a week for a duration of 6 months. These patients were examined for the signs of masseter muscular hypertrophy and pain in these muscles after palpation, wear in teeth and glare in dental restoration [34]. The study by Isa Kara et al. [31] also involved patients with probable SB. Apart from the medical interview gained from the patient or their bedroom partner/family members, the subjects had to fulfill at least one secondary clinical criterion to be included in the study.

### 5.3. The Study’s Strong and Weak Points

This systemic review provides a preliminary summary of current evidence regarding the association between SB and systemic inflammation. Results of the systematic review indicate that the levels of proinflammatory parameters can be elevated in subjects with SB. The conclusions obtained from this systematic review are preliminary due to a relatively small number of studies included and their heterogeneity. This systematic review indicated preliminary support for an association between SB and systemic inflammation; however, the inflammatory status was examined differently in each of the studies included in the review. Despite the evidence on the prognostic relationship between systemic inflammatory response markers and all-cause mortality [60], until now, no consensus has been established on the standard biomarkers to be used in analyzing the presence of health-damaging chronic inflammation. This moderate evidence, obtained in the synthesis of five studies, cannot be compared with findings of any other previous review, as this is the first paper investigating the topic.

Although the overall evidence suggesting that SB may be an autoimmune disease is insufficient, its significance should be highlighted. No systematic review, or meta-analysis, has ever been performed investigating the examined associations. Some of the relationships observed in this systematic review suggest that the diagnosis of SB may potentially form links with other inflammatory diseases. This in turn can have crucial consequences, particularly in patients with systemic autoimmune disease, who may have a substantially higher risk of developing CVD in comparison with the general population [61]. Interestingly, in the study performed by Ahlberg,
J. et al. [62] on a nationwide twin cohort, the risk of all-cause mortality among all participants (n = 12,040) was 40% higher in weekly bruxers than in non-bruxers. Surprisingly, the risk was no longer observed after the adjustment with all studied covariates. The conclusion coming from the study that ‘bruxism does not kill’ undoubtedly supports the current approach of bruxism being a behavior, rather than a disease [3]. However, this statement could be too hasty. Taking into account the possible association between SB, systemic inflammation and an elevated risk of developing CVD, we remain of the opinion that further research should investigate whether the diagnosis of SB has a direct impact on cardiovascular events regarded as primary outcomes. We should remember that SB is not a homogenic condition.

Our study may inform clinical practice. Obtained findings, indicating that SB could be associated with elevated levels of proinflammatory cytokines, encourage the expansion of health promotion programs. Taking into account that CVDs are the leading cause of death globally [63], broad target populations may be a vector in the successful implementation of health promotion programs [64]. The insufficient availability of sleep laboratories, where patients can undergo polysomnographic examination, makes a gap between the demand and supply, one of the main issues that public health in the field of sleep medicine is currently facing. The state of things does not become much more optimistic taking into account that the most reliable way of assessing the severity of SB, especially in patients who also report other sleep disturbances, continues to be camera-based polysomnography [65]. The findings of the study performed by Smardz, J. et al. [44] revealed that all the examined SB indices had significantly higher values for the recordings made without the use of a camera during polysomnography. According to the data, non-camera recordings had a diminished sensitivity and specificity in identifying SB including mild-to-moderate SB according to suggested criteria. A decreased sensitivity in diagnosing severe SB was also observed. The Standardised Tool for the Assessment of Bruxism (STAB) is likely to influence both clinical and research fields regarding the assessment of SB [66]. As of now, however, the STAB has not been widely accepted as a SB assessment tool, and the BruxScreen recently proposed by Lobbezoo et al. [67] could be used for the screening and overall psychometric testing of SB in clinical practice.

One of the limitations of the assessed studies is the inability to clarify the order in which SB and the increase in proinflammatory parameters occur. On the one hand, there have been models presented suggesting that the prevalence and intensity of SB can be positively correlated with psychological resilience [68]. However, on the other hand, it may also be assumed that the elevation of proinflammatory parameters is a result of cytokines being released into the bloodstream from the masseter muscle in response to tooth clenching. As of now, the question on whether the correlation between systemic inflammation and SB incidence is causal remains unanswered.

There are several other reasons for considering the present findings as preliminary. First, a small number of studies were entered into the systematic review. Second, the studied populations and indicators of SB and systemic inflammation were of a high heterogeneity. The EBSCOhost search engine was used to perform the search. It allowed for simultaneously screening different databases, out of which five were chosen. There are some well-recognized databases, like Scopus and Web of Science, that are not provided by EBSCOhost. Therefore, the results from these databases were screened based on the same keywords and no other relevant studies that could be included in the analysis were found.

Future studies should use more precise methods of assessing systemic inflammation. The precise assessment of the inflammatory status would allow for a better comparison between various studies. The limited number of studies did not allow for a meta-analysis to be conducted. Furthermore, other potential sociodemographic factors, such as gender, were not analyzed because the original studies did not provide data allowing for the calculation of SB–inflammation association coefficients for various subgroups. In fact, the patients in the studies performed by Louca Jounger et al. and Lago-Rizzardi et al. were only women [32,33]. This in turn reflects the fact that most findings underline the gender inequalities that exist in terms of TMD-related symptoms, to the disadvantage of women [69]. The link between psychosocial factors and pain in patients with painful TMD and different types of headaches [70] opens up a broad area of research needed in the field of pain medicine, as there is a premise of the correlation between self-reported pain and quality of life in patients with TMD [71]. The study performed by Karacay et al. [72] revealed the association between probable SB and pain-related TMD according to the Diagnostic Criteria for Temporomandibular Disorders (DC/TMD). According to the scoping review by Manfredini, D. and Lobbezoo, F., however, the relationship between SB and TMD is dependent on the assessment strategies that are adopted for SB [59]. In studies based on a low specificity for SB assessment (like a questionnaire or self-reported SB), there was a positive association with TMD pain. In contrast, studies utilizing electromyographic or polysomnographic examination found that the level of association between TMD pain and SB was much lower, at times even indicating a negative relationship between the two [59,73,74,75]. Recently, a critical evaluation of systematic reviews on the topic of therapies for SB in dentistry performed by Ceron et al. [76] showed that the results of the therapies were diverse and confusing. We still lack high-quality evidence on the most effective management of SB.

## 6. Conclusions

Whether SB can be considered a chronic inflammatory condition warrants further investigation. Despite its limitations, this systematic review provides a novel insight into the associations between SB and SCI. The systematic review findings, based on original studies enrolling patients suffering from tooth clenching, suggested that a higher intensity of SB could be associated with higher levels of proinflammatory parameters. However, each of the examined studies utilized different methods of assessing systemic inflammation, which makes the results dubious. The systemic response to SB as well as the causality between SB and the elevation of proinflammatory markers should be further investigated. The possible link between SB and cardiovascular risk makes it even more important.

## Figures and Tables

**Figure 1 brainsci-13-01104-f001:**
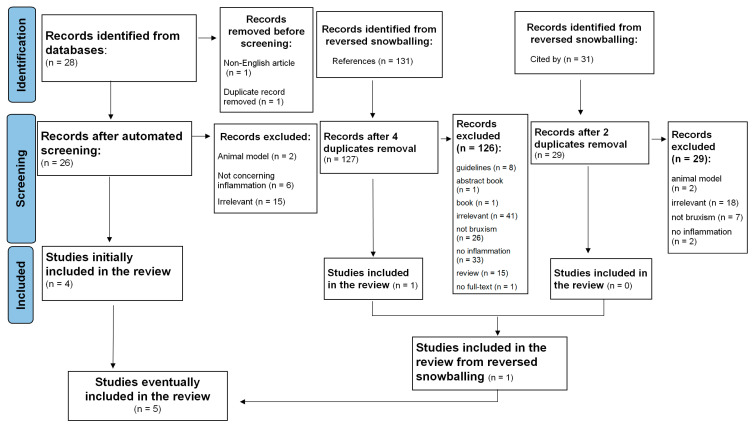
Flowchart of the systematic literature search according to PRISMA 2020 guidelines.

**Figure 2 brainsci-13-01104-f002:**
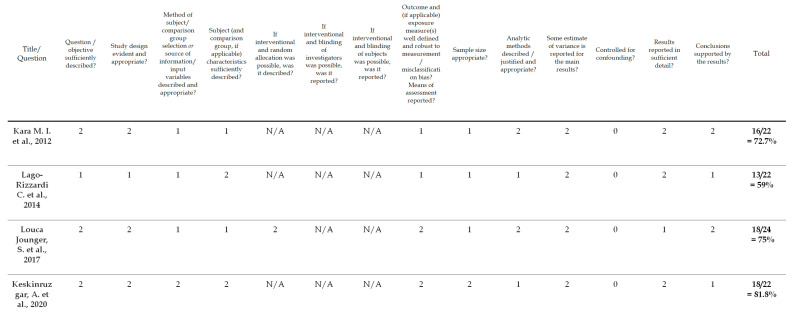
Risk of bias/checklist for quantitative studies [31,32,33,34].

**Figure 3 brainsci-13-01104-f003:**
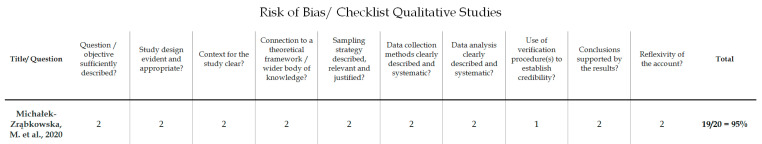
Risk of bias/checklist for qualitative studies [18].

**Table 1 brainsci-13-01104-t001:** Summary of the included studies.

Author and Year	Study Design	Population	Sample Size	Patients, n	Control, n	Females, n (%)	Mean Age, Years	Aim of Study	Inflammation Evaluation	Comments
Kara, M.I. et al., 2012 [31]	case–control study	SB diagnosed based on patient history and clinical criteria	65	33	32	33 (51)	20 [18,19,20,21,22,23]	evaluation of the relationship among SB and antioxidant/oxidant status to determine whether antioxidant/oxidant status may be used as a biological marker of SB	in 2 of the 3 assays performed, SB patients exhibited an oxidative imbalance	total oxidant status (TOS) levels were determined by using a novel automated colorimetric measurement method
Lago-Rizzardi, C. et al., 2014 [32]	case–control study	CFMP	48	24	24	48 (100)	45 ± 15	to investigate blood parameters associated with stress in patients with facial musculoskeletal pain	although not statistically different, CRP, IgE, C3, C4 and leukocytes were higher in the study group	patients enrolled for the study suffered from CFMP, not bruxism. However, bruxism appeared to be one of the main complaints—62.5% in the study group vs. 0.0% in the control (*p* < 0.001)
Louca Jounger, S. et al., 2017 [33]	case–control study	TMD	40	20	20	40 (100)	30 ± 10	to investigate cytokine levels in the masseter muscle, their response to experimental tooth clenching and their relation to pain, fatigue and psychological distress	the masseter levels of IL-6, IL-7, IL-8 and IL-13 were higher in TMD myalgia patients than controls and repetitive tooth clenching increased the levels of IL-6 and IL-8 in both groups, while IL-7, IL-13 and TNF increased only in patients	this study investigated inflammatory response in the muscle, not a systemic response. The patients enrolled for the study experienced temporomandibular disorders, not bruxism itself. However, clenching can potentially convey an effect similar to what we see in bruxism
Keskinruzgar, A. et al., 2020 [34]	case–control study	SB diagnosed based on patient history and clinical criteria	80	40	40	46 (58)	29 ± 9	to evaluate the anxiety and OCT findings in patients with SB and to develop objective measurements to assess the neurodegenerative and inflammatory processes associated with this disease	the measurements of RNFL, IPL and GCL in the bruxism group were significantly lower than the control group, whereas the choroidal thickness was significantly higher	this study’s results showed that SB is a neurodegenerative and inflammatory process
Michałek-Zrąbkowska, M. et al., 2020 [18]	observational, prospective cohort study	SB diagnosed based on AASM standards	74	74	-	54 (73)	34 ± 11	to determine inflammatory markers in individuals with SB	the bruxism episode index (BEI) positively correlated with the concentrations of 17-hydroxycorticosteroids and C-reactive protein	none

SB—sleep bruxism, CFMP—chronic facial musculoskeletal pain, TMD—temporomandibular disorders, RNFL—retinal nerve fiber layer, IPL—inner plexiform layer, GCL—ganglion cell layer, OCT—optical coherence tomography, AASM—American Academy of Sleep Medicine, CRP—C-reactive protein, IgE—immunoglobulin E, C3—complement C3, C4—complement C4, Il-6—interleukin 6, Il-7—interleukin 7, Il-8—interleukin 8, Il-13—interleukin 13, TNF—tumor necrosis factor.

## Data Availability

The data presented in this study are available upon request from thecorresponding author. The data are not publicly available.

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
