# Peer review of "Systematic Review on the Link between Sleep Bruxism and Systemic Chronic Inflammation"

_brainsci, 2023, doi:10.3390/brainsci13071104_

Round 1
Reviewer 1 Report
This is a well-written manuscript. Due to the small number of relevant studies and the heterogeneity of the studies, the association between sleep bruxism and systemic chronic inflammation remain ambiguous despite this systematic review. Nevertheless, I agree that this study can provides a novel insight into the associations between them.
Minor comments
1. The abstract relatively too emphasizes the methodological part of the systematic review. It should include more about the results of the study.
2. Figure 1 needs to be improved for readability. Box size can be made larger and the size of the text inside the box also can be increased.
3. If an abbreviation of the full spelling is already given, the word that follows should be written as an abbreviation. However, it is often observed that it is not the case. For example, systemic chronic inflammation in line 56, sleep bruxism in line 59, and sleep bruxism in line 112, etc.
Author Response
This is a well-written manuscript. Due to the small number of relevant studies and the heterogeneity of the studies, the association between sleep bruxism and systemic chronic inflammation remain ambiguous despite this systematic review. Nevertheless, I agree that this study can provides a novel insight into the associations between them.
Response: We appreciate the reviewer’s comment. We agree that there is a small number of studies on this topic, however taken into consideration that systemic chronic inflammation has been found to increase the cardiovascular risk ultimately reducing the duration of healthy life years, we aim to raise this topic and potentially show an indication for future studies.
-
The abstract relatively too emphasizes the methodological part of the systematic review. It should include more about the results of the study.
Response: We would like to thank the reviewer for this feedback. We have modified the abstract accordingly, to emphasize that this systematic review revealed there are actually no papers suggesting that the inflammatory status in bruxers is comparable to non-bruxers.
-
Figure 1 needs to be improved for readability. Box size can be made larger and the size of the text inside the box also can be increased.
Response: We agree with the reviewer’s statement and therefore we have modified the figure 1 so that it is more readable. Now the size of the text is increased and we believe that through this modification the clarity of our manuscript has risen. We appreciate pointing this out.
-
If an abbreviation of the full spelling is already given, the word that follows should be written as an abbreviation. However, it is often observed that it is not the case. For example, systemic chronic inflammation in line 56, sleep bruxism in line 59, and sleep bruxism in line 112, etc.
Response: Thank you. We have applied appropriate modifications and we have implemented the abbreviations after an abbreviation of the full spelling was already given.
Reviewer 2 Report
Title: it is missing a letter "s" in "systemic"
Introduction: I would remove the firts phrase, it was lost. Another point to highlight is there was not a raciocine line from cause to consequence. Secondary titles were also unecessaire.
Methods: the ending of introduction is actually the begginging of methods. Please, change it. The main point that worries me is the fact that the search in the databases gave much less than the manual search. This means that your search strategy was neither sensitive nor specific and this is a very serious bias! It is also not necessary for the two reviewers to perform the search, but it is extremely necessary to improve the search. In addition, I missed some databases, such as scopus, web of science. In elegibility criteria, how did you diagnose systemic chronic infammation? Were are the information about qaulitative/quantitative analysis?
Results: You mentioned PRISMA in abstract, however many PRISMA topics are missing in the text. Review it, please. The beggining of result section is too much confuse! You must start relating how many articles you found after searching in database (this info was found in "methods"...). Where are the result of risk of bias? The study of "Louca Jounger S. et al." did not mentioned bruxism, so it must be excluded.
My greatest understanding is in bruxism and in the systematic review methodology. About bruxism, I found the information confusing, as well as the way in which to select the included articles. Regarding systematic reviews, many points need to be improved, which may change the result. Therefore, I await these changes to better analyze.
The English was good
Author Response
Title: it is missing a letter "s" in "systemic"
Response: Thank you, that was an oversight. The title has been corrected.
Introduction: I would remove the firts phrase, it was lost. Another point to highlight is there was not a raciocine line from cause to consequence.
Response: We thank the reviewer for this comment. The challenge to find the causal relationship between chronic inflammation and sleep bruxism is crucial and we agree that there has not been provided a ratiocinate on the cause and consequence between them, but there is still no data available on that topic and therefore we have focused on the co-existence of higher inflammatory status and sleep bruxism. We have added according statement in the introduction.
Secondary titles were also unecessaire.
Response: We appreciate that the reviewer pointed this out. Accordingly to the reviewer’s suggestion we have removed the secondary title “The clinical relevance of the topic” as it still is a part of the introduction . However, the subtitles in our work result from the requirements of Prisma. We believe that through this modification our manuscript is more clear and we would like to thank the reviewer for this noteworthy comment.
Methods: the ending of introduction is actually the begginging of methods. Please, change it
Response: Thank you for the comment. We have modified the ending of the introduction section accordingly: “Up to this point, there has not been proven causal relationship between SB and SCI. We still lack evidence to propose cause to consequence ratiocinate and therefore, given the high prevalence and potentially significant consequences of SB coexisting with systemic inflammation, we aimed to establish the current state of knowledge on that relationship.” In the methods section we have precisely described the combination of keywords used in the search, the databases as well as the process of reverse-snowballing method.
The main point that worries me is the fact that the search in the databases gave much less than the manual search. This means that your search strategy was neither sensitive nor specific and this is a very serious bias! It is also not necessary for the two reviewers to perform the search, but it is extremely necessary to improve the search.
Response: The search in Medline database using keywords “sleep bruxism” AND “inflammation” results in only 3 results, none of which is suitable to include into the analysis. Through the profound manual search none of relevant studies that should be included in the analysis other than the ones already included have been found. Therefore we cannot entirely agree with the reviewer’s statement that the search in the databases gave much less than the manual search.
In addition, I missed some databases, such as scopus, web of science.
Response: To perform the search we have used the EBSCOhost search engine. It allows to screen simultaneously different databases, out of which we have selected MEDLINE, Health Source—Consumer Edition, Health Source: Nursing/Academic Edition, Academic Search Ultimate and ERIC. We agree with the reviewer’s statement that the above mentioned databases – Scopus and Web of science are well recognized databases. However, we have gone through the results from those databases basing on the same key words and no other relevant studies that could be included in the analysis have been found. We have also added appropriate justification in discussion.
After the selection of five databases in EBSCOhost Web using the following combination of keywords: (‘sleep bruxism’ OR ‘bruxing’ OR ‘bruxer*’) AND (‘systemic inflammation’ OR ‘inflammation*’ OR ‘inflammatory response’ OR ‘IL-1’ OR ‘Interleukin-1’ OR ‘Il-6’ OR ‘Interleukin-6’ OR ‘Il-8’ OR ‘Interleukin-8’ OR ‘cytokine*’ OR ‘CRP’ OR ‘C-reactive protein’ OR ‘TNF-alpha’ OR ‘TNF-α’ OR ‘fibrinogen’) we have received 28 records. Thanks to using EBSCOhost Web there was no need to perform the search 5 times separately. EBSCOhost web has provided the results from 5 different databases, the strategy is similar to what can be found in those two articles: DOI: 10.1007/s00392-023-02256-7 and DOI: 10.3390/jcm11195519
In elegibility criteria, how did you diagnose systemic chronic infammation? Were are the information about qaulitative/quantitative analysis?
Response: We support the reviewer’s statement that the determination in eligibility criteria of how systemic chronic inflammation is diagnosed would elevate the credibility of the study. Taken into account the small numbers of studies concerning this topic, we could not have been strict in terms of eligibility criteria. This systematic review has indicated preliminary support for an association between SB and systemic inflammation, however the inflammatory status was examined differently in each of the studies included in the review. Undoubtedly, our review is a clear indicator that future studies should use more precise methods of assessing systemic inflammation. Precise assessment of the inflammatory status would allow for a better comparison between those studies.
Results: You mentioned PRISMA in abstract, however many PRISMA topics are missing in the text. Review it, please.
Response: The reviewer is right, there are some points in PRISMA that are missing. It is the result of small number of studies included in the analysis, as well as the heterogeneity between them. Different ways of assessing the inflammatory status in the articles disables performing a comparison head to head.
beggining of result section is too much confuse! You must start relating how many articles you found after searching in database (this info was found in "methods"...).
Response: We appreciate this valuable comment. We have improved the result section with adding following paragraph: “The results of the search are displayed in Fig. 1. The research revealed 5 articles, in-cluding 307 participants—191 patients and 116 controls. Four studies were quantitative studies and one study was a qualitative study. The characteristic of the included studies is described in Figure 1”. We would like to thank the reviewer for pointing this out, as we believe that it made this section more clear to future readers.
Where are the result of risk of bias?
Response: We have added the figures presenting the results of risk of bias. Please see it below:
The study of "Louca Jounger S. et al." did not mentioned bruxism, so it must be excluded.
Response: We agree with the reviewer, that in the study by Louca Jounger S. et al. the bruxism has not been mentioned itself. However the intervention of tooth-clenching can potentially convey the effect similar to what we see in bruxism, as tooth-clenching is a primal symptom of bruxism according to the definition of bruxism as: “a repetitive masticatory muscle activity characterised by clenching or grinding of the teeth and/or by bracing or thrusting of the mandible and specified as either sleep bruxism or awake bruxism”.
Lobbezoo F, Ahlberg J, Glaros AG, Kato T, Koyano K, Lavigne GJ, de Leeuw R, Manfredini D, Svensson P, Winocur E. Bruxism defined and graded: an international consensus. J Oral Rehabil. 2013 Jan;40(1):2-4. doi: 10.1111/joor.12011. Epub 2012 Nov 4. PMID: 23121262.
Thus, tooth clenching described in that paper is typical activity of bruxism. Taken into consideration important results, as it shows that repetitive tooth-clenching increases the levels of inflammatory markers, we would prefer to not to exclude the study by Louca Jounger S. et al.
My greatest understanding is in bruxism and in the systematic review methodology. About bruxism, I found the information confusing, as well as the way in which to select the included articles. Regarding systematic reviews, many points need to be improved, which may change the result. Therefore, I await these changes to better analyze.
Response: Thank you for your time and valuable comments. We have improved the manuscript according to the all reviewer's comments and suggestions. We believe that implementing the suggestions contributed to making the article better.
